# Learning with Feature Evolvable Streams

**Bo-Jian Hou**    **Lijun Zhang**    **Zhi-Hua Zhou**
National Key Laboratory for Novel Software Technology,
Nanjing University, Nanjing, 210023, China
`{houbj,zhanglj,zhouzh}@lamda.nju.edu.cn`

## Abstract

Learning with streaming data has attracted much attention during the past few years. Though most studies consider data stream with fixed features, in real practice the features may be evolvable. For example, features of data gathered by limited-lifespan sensors will change when these sensors are substituted by new ones. In this paper, we propose a novel learning paradigm: *Feature Evolvable Streaming Learning* where old features would vanish and new features would occur. Rather than relying on only the current features, we attempt to recover the vanished features and exploit it to improve performance. Specifically, we learn two models from the recovered features and the current features, respectively. To benefit from the recovered features, we develop two ensemble methods. In the first method, we combine the predictions from two models and theoretically show that with the assistance of old features, the performance on new features can be improved. In the second approach, we dynamically select the best single prediction and establish a better performance guarantee when the best model switches. Experiments on both synthetic and real data validate the effectiveness of our proposal.

## 1 Introduction

In many real tasks, data are accumulated over time, and thus, learning with streaming data has attracted much attention during the past few years. Many effective approaches have been developed, such as hoeffding tree [7], Bayes tree [27], evolving granular neural network (eGNN) [17], Core Vector Machine (CVM) [29], etc. Though these approaches are effective for certain scenarios, they have a common assumption, i.e., the data stream comes with a fixed stable feature space. In other words, the data samples are always described by the same set of features. Unfortunately, this assumption does not hold in many streaming tasks. For example, for ecosystem protection one can deploy many sensors in a reserve to collect data, where each sensor corresponds to an attribute/feature. Due to its limited-lifespan, after some periods many sensors will wear out, whereas some new sensors can be spread. Thus, features corresponding to the old sensors vanish while features corresponding to the new sensors appear, and the learning algorithm needs to work well under such evolving environment. Note that the ability of adapting to environmental change is one of the fundamental requirements for *learnware* [37], where an important aspect is the ability of handling evolvable features.

A straightforward approach is to rely on the new features and learn a new model to use. However, this solution suffers from some deficiencies. First, when new features just emerge, there are few data samples described by these features, and thus, the training samples might be insufficient to train a strong model. Second, the old model of vanished features is ignored, which is a big waste of our data collection effort. To address these limitations, in this paper we propose a novel learning paradigm: *Feature Evolvable Streaming Learning* (FESL). We formulate the problem based on a key observation: in general features do not change in an arbitrary way; instead, there are some overlapping periods in which both old and new features are available. Back to the ecosystem protection example, since the lifespan of sensors is known to us, e.g., how long their battery will run out is a prior knowledge, we

usually spread a set of new sensors before the old ones wear out. Thus, the data stream arrives in a way as shown in Figure 1, where in period $T_1$, the original set of features are valid and at the end of $T_1$, period $B_1$ appears, where the original set of features are still accessible, but some new features are included; then in $T_2$, the original set of features vanish, only the new features are valid but at the end of $T_2$, period $B_2$ appears where newer features come. This process will repeat again and again. Note that the $T_1$ and $T_2$ periods are usually long, whereas the $B_1$ and $B_2$ periods are short because, as in the ecosystem protection example, the $B_1$ and $B_2$ periods are just used to switch the sensors and we do not want to waste a lot of lifetime of sensors for such overlapping periods.

In this paper, we propose to solve the FESL problem by utilizing the overlapping period to discover the relationship between the old and new features, and exploiting the old model even when *only* the new features are available. Specifically, we try to learn a mapping from new features to old features through the samples in the overlapping period. In this way, we are able to reconstruct old features from new ones and thus the old model can still be applied. To benefit from additional features, we develop two ensemble methods, one is in a combination manner and the other in a dynamic selection manner. In

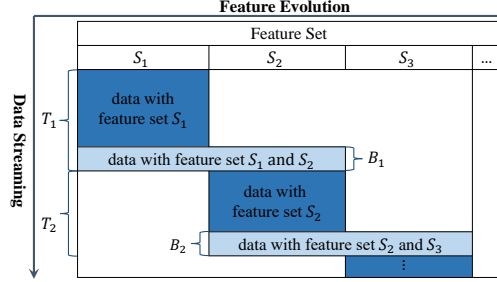

Figure 1: Illustration that how data stream comes.

the first method, we combine the predictions from two models and theoretically show that with the assistance of old features, the performance on new features can be improved. In the second approach, we dynamically select the best single prediction and establish a better performance guarantee when the best model switches at an arbitrary time. Experiments on synthetic and real datasets validate the effectiveness of our proposal.

The rest of this paper is organized as follows. Section 2 introduces related work. Section 3 presents the formulation of FESL. Our proposed approaches with corresponding analyses are presented in section 4. Section 5 reports experimental results. Finally, Section 6 concludes.

## 2    Related Work

Data stream mining contains several tasks, including classification, clustering, frequency counting, and time series analysis. Our work is most related to the classification task and we can also solve the regression problem. Existing techniques for data stream classification can be divided into two categories, one only considers a single classifier and the other considers ensemble classifiers. For the former, several methods origin from approaches such as decision tree [7], Bayesian classification [27], neural networks [17], support vector machines [29], and $k$-nearest neighbour [1]. For the latter, various ensemble methods have been proposed including Online Bagging & Boosting [22], Weighted Ensemble Classifiers [30, 20], Adapted One-vs-All Decision Trees (OVA) [12] and Meta-knowledge Ensemble [33]. For more details, please refer to [9, 10, 2, 6, 21]. These traditional streaming data algorithms often assume that the data samples are described by the same set of features, while in many real streaming tasks feature often changes. We want to emphasize that though concept-drift happens in streaming data where the underlying data distribution changes over time [2, 10, 4], the number of features in concept-drift never changes which is different from our problem. Most studies correlated to features changing are focusing on feature selection and extraction [26, 35] and to the best of our knowledge, none of them consider the evolving of feature set during the learning process.

Data stream mining is a hot research direction in the area of data mining while online learning [38, 14] is a related topic from the area of machine learning. Yet online learning can also tackle the streaming data problem since it assumes that the data come in a streaming way. Online learning has been extensively studied under different settings, such as learning with experts [5] and online convex optimization [13, 28]. There are strong theoretical guarantees for online learning, and it usually uses regret or the number of mistakes to measure the performance of the learning procedure. However, most of existing online learning algorithms are limited to the case that the feature set is fixed. Other related topics involving multiple feature sets include multi-view learning [18, 19, 32], transfer learning [23, 24] and incremental attribute learning [11]. Although both our approaches and multi-view learning exploit the relation between different sets of features, there exists a fundamental

difference: multi-view learning assumes that every sample is described by multiple feature sets simultaneously, whereas in FESL only few samples in the feature switching period have two sets of features, and no matter how many periods there are, the switching part involves only two sets of features. Transfer learning usually assumes that data are in batch mode, few of them consider the streaming cases where data arrives sequentially and cannot be stored completely. One exception is online transfer learning [34] in which data from both sets of features arrive sequentially. However, they assume that all the feature spaces must appear simultaneously during the whole learning process while such an assumption is not available in FESL. When it comes to incremental attribute learning, old sets of features do not vanish or do not vanish entirely while in FESL, old ones will vanish thoroughly when new sets of features come.

The most related work is [15], which also handles evolving features in streaming data. Different to our setting where there are overlapping periods, [15] handles situations where there is no overlapping period but there are overlapping features. Thus, the technical challenges and solutions are different.

## 3  Preliminaries

We focus on both classification and regression tasks. On each round of the learning process, the algorithm observes an instance and gives its prediction. After the prediction has been made, the true label is revealed and the algorithm suffers a *loss* which reflects the discrepancy between the prediction and the groundtruth. We define "feature space" in our paper by a set of features. That the feature space changes means both the underlying distribution of the feature set and the number of features change. Consider the process with three periods where in the first period large amount of data stream come from the old feature space; then in the second period named as overlapping period, few of data come from both the old and the new feature space; soon afterwards in the third period, data stream only

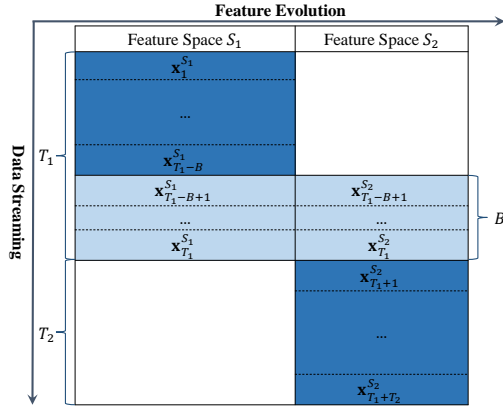

Figure 2: Specific illustration with one cycle.

come from the new feature space. We call this whole process a cycle. As can be seen from Figure 1, each cycle merely includes two feature spaces. Thus, we only need to focus on one cycle and it is easy to extend to the case with multiple cycles. Besides, we assume that the old features in one cycle will vanish simultaneously by considering the example that in ecosystem protection, all the sensors share the same expected lifespan and thus they will wear out at the same time. We will study the case where old features do not vanish simultaneously in the future work.

Based on the above discussion, we only consider two feature spaces denoted by $S_1$ and $S_2$, respectively. Suppose that in the overlapping period, there are $B$ rounds of instances both from $S_1$ and $S_2$. As can be seen from Figure 2, the process can be concluded as follows.

- For $t = 1, \ldots, T_1 - B$, in each round, the learner observes a vector $\mathbf{x}_t^{S_1} \in \mathbb{R}^{d_1}$ sampled from $S_1$ where $d_1$ is the number of features of $S_1$, $T_1$ is the number of total rounds in $S_1$.
- For $t = T_1 - B + 1, \ldots, T_1$, in each round, the learner observes two vectors $\mathbf{x}_t^{S_1} \in \mathbb{R}^{d_1}$ and $\mathbf{x}_t^{S_2} \in \mathbb{R}^{d_2}$ from $S_1$ and $S_2$, respectively where $d_2$ is the number of features of $S_2$.
- For $t = T_1 + 1, \ldots, T_1 + T_2$, in each round, the learner observes a vector $\mathbf{x}_t^{S_2} \in \mathbb{R}^{d_2}$ sampled from $S_2$ where $T_2$ is the number of rounds in $S_2$. Note that $B$ is small, so we can omit the streaming data from $S_2$ on rounds $T_1 - B + 1, \ldots, T_1$ since they have minor effect on training the model in $S_2$.

We use $\|\mathbf{x}\|$ to denote the $\ell_2$-norm of a vector $\mathbf{x} \in \mathbb{R}^{d_i}$, $i = 1, 2$. The inner product is denoted by $\langle \cdot, \cdot \rangle$. Let $\Omega_1 \subseteq \mathbb{R}^{d_1}$ and $\Omega_2 \subseteq \mathbb{R}^{d_2}$ be two sets of linear models that we are interested in. We define the projection $\Pi_{\Omega_i}(\mathbf{b}) = \operatorname{argmin}_{\mathbf{a} \in \Omega_i} \|\mathbf{a} - \mathbf{b}\|$, $i = 1, 2$. We restrict our prediction function in $i$-th feature space and $t$-th round to be linear which takes the form $\langle \mathbf{w}_{i,t}, \mathbf{x}_t^{S_i} \rangle$ where $\mathbf{w}_{i,t} \in \mathbb{R}^{d_i}$, $i = 1, 2$. The loss function $\ell(\mathbf{w}^\top \mathbf{x}, y)$ is convex in its first argument and in implementing algorithms, we use

---
**Algorithm 1** Initialize
---
1: Initialize $\mathbf{w}_{1,1} \in \Omega_1$ randomly, $M_1 = 0$, and $M_2 = 0$;
2: **for** $t = 1, 2, \ldots, T_1$ **do**
3:      Receive $\mathbf{x}_t^{S_1} \in \mathbb{R}^{d_1}$ and predict $f_t = \mathbf{w}_{1,t}^\top \mathbf{x}_t^{S_1} \in \mathbb{R}$; Receive the target $y_t \in \mathbb{R}$, and suffer loss $\ell(f_t, y_t)$;
4:      Update $\mathbf{w}_{1,t}$ using (1) where $\tau_t = 1/\sqrt{t}$;
5:      **if** $t > T_1 - B$ **then** $M_1 = M_1 + \mathbf{x}_t^{S_2} \mathbf{x}_t^{S_2 \top}$ and $M_2 = M_2 + \mathbf{x}_t^{S_2} \mathbf{x}_t^{S_1 \top}$;
6: $\boldsymbol{M}_* = M_1^{-1} M_2$.
---

*logistic loss* for classification task, namely $\ell(\mathbf{w}^\top \mathbf{x}, y) = (1/\ln 2) \ln(1 + \exp(-y(\mathbf{w}^\top \mathbf{x})))$ and *square loss* for regression task, namely $\ell(\mathbf{w}^\top \mathbf{x}, y) = (y - \mathbf{w}^\top \mathbf{x})^2$.

The most straightforward or baseline algorithm is to apply online gradient descent [38] on rounds $1, \ldots, T_1$ with streaming data $\mathbf{x}_t^{S_1}$, and invoke it again on rounds $T_1 + 1, \ldots, T_1 + T_2$ with streaming data $\mathbf{x}_t^{S_2}$. The models are updated according to (1), where $\tau_t$ is a varied step size:

$$\mathbf{w}_{i,t+1} = \Pi_{\Omega_i} \left( \mathbf{w}_{i,t} - \tau_t \nabla \ell(\mathbf{w}_{i,t}^\top \mathbf{x}_t^{S_i}, y_t) \right), \; i = 1, 2. \tag{1}$$

## 4 Our Proposed Approach

In this section, we first introduce the basic idea of the solution to FESL, then two different kinds of approaches with the corresponding analyses are proposed.

The major limitation of the baseline algorithm mentioned above is that the model learned on rounds $1, \ldots, T_1$ is ignored on rounds $T_1 + 1, \ldots, T_1 + T_2$. The reason is that from rounds $t > T_1$, we cannot observe data from feature space $S_1$, and thus the model $\mathbf{w}_{1,T_1}$, which operates in $S_1$, cannot be used directly. To address this challenge, we assume there is a certain relationship $\psi : \mathbb{R}^{d_2} \to \mathbb{R}^{d_1}$ between the two feature spaces, and we try to discover it in the overlapping period. There are several methods to learn a relationship between two sets of features including multivariate regression [16], streaming multi-label learning [25], etc. In our setting, since the overlapping period is very short, it is unrealistic to learn a complex relationship between the two spaces. Instead, we use a linear mapping to approximate $\psi$. Assume the coefficient matrix of the linear mapping is $M$, then during rounds $T_1 - B + 1, \ldots, T_1$, the estimation of $\boldsymbol{M}$ can be based on least squares

$$\min_{\boldsymbol{M} \in \mathbb{R}^{d_2 \times d_1}} \sum_{t=T_1-B+1}^{T_1} \|\mathbf{x}_t^{S_1} - \boldsymbol{M}^\top \mathbf{x}_t^{S_2}\|_2^2.$$

The optimal solution $\boldsymbol{M}_*$ to the above problem is given by

$$\boldsymbol{M}_* = \left( \sum_{t=T_1-B+1}^{T_1} \mathbf{x}_t^{S_2} \mathbf{x}_t^{S_2 \top} \right)^{-1} \left( \sum_{t=T_1-B+1}^{T_1} \mathbf{x}_t^{S_2} \mathbf{x}_t^{S_1 \top} \right).$$

Then if we only observe an instance $\mathbf{x}_t^{S_2} \in \mathbb{R}^{d_2}$ from $S_2$, we can recover an instance in $S_1$ by $\psi(\mathbf{x}^{S_2}) \in \mathbb{R}^{d_1}$, to which $\mathbf{w}_{1,T_1}$ can be applied. Based on this idea, we will make two changes to the baseline algorithm:

- During rounds $T_1 - B + 1, \ldots, T_1$, we will learn a relationship $\psi$ from $(\mathbf{x}_{T_1-B+1}^{S_1}, \mathbf{x}_{T_1-B+1}^{S_2})$, $\ldots, (\mathbf{x}_{T_1}^{S_1}, \mathbf{x}_{T_1}^{S_2})$.
- From rounds $t > T_1$, we will keep on updating $\mathbf{w}_{1,t}$ using the recovered data $\psi(\mathbf{x}_t^{S_2})$ and predict the target by utilizing the predictions of $\mathbf{w}_{1,t}$ and $\mathbf{w}_{2,t}$.

In round $t > T_1$, the learner can calculate two base predictions based on models $\mathbf{w}_{1,t}$ and $\mathbf{w}_{2,t}$: $f_{1,t} = \mathbf{w}_{1,t}^\top(\psi(\mathbf{x}_t^{S_2}))$ and $f_{2,t} = \mathbf{w}_{2,t}^\top \mathbf{x}_t^{S_2}$. By utilizing the two base predictions in each round, we propose two methods, both of which are able to follow the better base prediction empirically and theoretically. The process to obtain the relationship mapping $\psi$ and $\mathbf{w}_{1,T_1}$ during rounds $1, \ldots, T_1$ are concluded in Algorithm 1.

---

**Algorithm 2** FESL-c(ombination)

1: Initialize $\psi$ and $\mathbf{w}_{1,T_1}$ during $1, \ldots, T_1$ using Algorithm 1;
2: $\alpha_{1,T_1} = \alpha_{2,T_1} = \frac{1}{2}$;
3: Initialize $\mathbf{w}_{2,T_1+1}$ randomly and $\mathbf{w}_{1,T_1+1}$ by $\mathbf{w}_{1,T_1}$;
4: **for** $t = T_1 + 1, T_1 + 2, \ldots, T_1 + T_2$ **do**
5: 　　Receive $\mathbf{x}_t^{S_2} \in \mathbb{R}^{S_2}$ and predict $f_{1,t} = \mathbf{w}_{1,t}^\top(\psi(\mathbf{x}_t^{S_2}))$ and $f_{2,t} = \mathbf{w}_{2,t}^\top \mathbf{x}_t^{S_2}$;
6: 　　Predict $\widehat{p}_t \in \mathbb{R}$ using (2), then receive the target $y_t \in \mathbb{R}$, and suffer loss $\ell(\widehat{p}_t, y_t)$;
7: 　　Update weights using (3) where $\eta = \sqrt{8(\ln 2)/T_2}$;
8: 　　Update $\mathbf{w}_{1,t}$ and $\mathbf{w}_{2,t}$ using (4) and (1) respectively where $\tau_t = 1/\sqrt{t - T_1}$;

---

## 4.1 Weighted Combination

We first propose an ensemble method by combining predictions with weights based on exponential of the cumulative loss [5]. The prediction at time $t$ is the weighted average of all the base predictions:

$$\widehat{p}_t = \alpha_{1,t} f_{1,t} + \alpha_{2,t} f_{2,t} \tag{2}$$

where $\alpha_{i,t}$ is the weight of the $i$-th base prediction. With the previous loss of each base model, we can update the weights of the two base models as follows:

$$\alpha_{i,t+1} = \frac{\alpha_{i,t} e^{-\eta \ell(f_{i,t}, y_t)}}{\sum_{j=1}^2 \alpha_{j,t} e^{-\eta \ell(f_{j,t}, y_t)}}, \; i = 1, 2, \tag{3}$$

where $\eta$ is a tuned parameter. The updating rule of the weights shows that if the loss of one of the models on previous round is large, then its weight will decrease in an exponential rate in next round, which is reasonable and can derive a good theoretical result shown in Theorem 1. Algorithm 2 summarizes our first approach for FESL named as FESL-c(ombination). We first learn a model $\mathbf{w}_{1,T_1}$ using online gradient descent on rounds $1, \ldots, T_1$, during which, we also learn a relationship $\psi$ for $t = T_1 - B + 1, \ldots, T_1$. For $t = T_1 + 1, \ldots, T_1 + T_2$, we learn a model $\mathbf{w}_{2,t}$ on each round and keep updating $\mathbf{w}_{1,t}$ on the recovered data $\psi(\mathbf{x}_t^{S_2})$ showed in (4) where $\tau_t$ is a varied step size:

$$\mathbf{w}_{1,t+1} = \Pi_{\Omega_i}\left(\mathbf{w}_{1,t} - \tau_t \nabla\ell(\mathbf{w}_{1,t}^\top(\psi(\mathbf{x}_t^{S_2})), y_t)\right). \tag{4}$$

Then we combine the predictions of the two models by weights calculated in (3).

**Analysis** In this paragraph, we borrow the *regret* from online learning to measure the performance of FESL-c. Specifically, we give a loss bound as follows which shows that the performance will be improved with assistance of the old feature space. For the sake of soundness, we put the proof of our theorems in the supplementary file. We define that $L^{S_1}$ and $L^{S_2}$ are two cumulative losses suffered by base models on rounds $T_1 + 1, \ldots, T_1 + T_2$,

$$L^{S_1} = \sum_{t=T_1+1}^{T_1+T_2} \ell(f_{1,t}, y_t), \; L^{S_2} = \sum_{t=T_1+1}^{T_1+T_2} \ell(f_{2,t}, y_t), \tag{5}$$

and $L^{S_{12}}$ is the cumulative loss suffered by our methods: $L^{S_{12}} = \sum_{t=T_1+1}^{T_1+T_2} \ell(\widehat{p}_t, y_t)$. Then we have:

**Theorem 1.** *Assume that the loss function $\ell$ is convex in its first argument and that it takes value in [0,1]. For all $T_2 > 1$ and for all $y_t \in \mathcal{Y}$ with $t = T_1 + 1, \ldots, T_1 + T_2$, $L^{S_{12}}$ with parameter $\eta_t = \sqrt{8(\ln 2)/T_2}$ satisfies*

$$L^{S_{12}} \le \min(L^{S_1}, L^{S_2}) + \sqrt{(T_2/2)\ln 2} \tag{6}$$

This theorem implies that the cumulative loss $L^{S_{12}}$ of Algorithm 2 over rounds $T_1 + 1, \ldots, T_1 + T_2$ is comparable to the minimum of $L^{S_1}$ and $L^{S_2}$. Furthermore, we define $C = \sqrt{(T_2/2)\ln 2}$. If $L^{S_2} - L^{S_1} > C$, it is easy to verify that $L^{S_{12}}$ is smaller than $L^{S_2}$. In summary, on rounds $T_1 + 1, \ldots, T_1 + T_2$, when $\mathbf{w}_{1,t}$ is better than $\mathbf{w}_{2,t}$ to certain degree, the model with assistance from $S_1$ is better than that without assistance.

---
**Algorithm 3** FESL-s(election)
---
1: Initialize $\psi$ and $\mathbf{w}_{1,T_1}$ during $1, \ldots, T_1$ using Algorithm 1;
2: $\alpha_{1,T_1} = \alpha_{2,T_1} = \frac{1}{2}$;
3: Initialize $\mathbf{w}_{2,T_1+1}$ randomly and $\mathbf{w}_{1,T_1+1}$ by $\mathbf{w}_{1,T_1}$;
4: **for** $t = T_1 + 1, T_1 + 2, \ldots, T_1 + T_2$ **do**
5:   Receive $\mathbf{x}_t^{S_2} \in \mathbb{R}^{S_2}$ and predict $f_{1,t} = \mathbf{w}_{1,t}^\top (\psi(\mathbf{x}_t^{S_2}))$ and $f_{2,t} = \mathbf{w}_{2,t}^\top \mathbf{x}_t^{S_2}$;
6:   Draw a model $\mathbf{w}_{i,t}$ according to the distribution (7) and predict $\widehat{p}_t = f_{i,t}$ according to the model;
7:   Receive the target $y_t \in \mathbb{R}$, and suffer loss $\ell(\widehat{p}_t, y_t)$; Update the weights using (8);
8:   Update $\mathbf{w}_{1,t}$ and $\mathbf{w}_{2,t}$ using (4) and (1) respectively, where $\tau_t = 1/\sqrt{t - T_1}$.
---

## 4.2 Dynamic Selection

The combination approach mentioned in the above subsection combines several base models to improve the overall performance. Generally, combination of several classifiers performs better than selecting only one single classifier [36]. However, it requires that the performance of base models should not be too bad, for example, in Adaboost the accuracy of the base classifiers should be no less than 0.5 [8]. Nevertheless, in our FESL problem, on rounds $T_1 + 1, \ldots, T_1 + T_2$, $\mathbf{w}_{2,t}$ cannot satisfy the requirement in the beginning due to insufficient training data and $\mathbf{w}_{1,t}$ may become worse when more and more data come causing a cumulation of recovered error. Thus, it may not be appropriate to combine the two models all the time, whereas dynamically selecting the best single may be a better choice. Hence we propose a method based on a new strategy, i.e., dynamic selection, similar to the Dynamic Classifier Selection [36] which only uses the best single model rather than combining both of them in each round. Note that, though we only select one of the models, we retain and utilize both of them to update their weights. So it is still an ensemble method. The basic idea of dynamic selection is to select the model of larger weight with higher probability. Algorithm 3 summarizes our second approach for FESL named as FESL-s(election). Specifically, the steps in Algorithm 3 on rounds $1, \ldots, T_1$ is the same as that in Algorithm 2. For $t = T_1 + 1, \ldots, T_1 + T_2$, we still update weights of each model. However, when doing prediction, we do not combine all the models' prediction, we adopt the result of the "best" model's according to the distribution of their weights

$$p_{i,t} = \frac{\alpha_{i,t-1}}{\sum_{j=1}^2 \alpha_{j,t-1}} \quad i = 1, 2. \tag{7}$$

To track the best model, we have a different way of updating weights which is given as follows [5].

$$v_{i,t} = \alpha_{i,t-1} e^{-\eta \ell(f_{i,t}, y_t)}, \ i = 1, 2, \quad \alpha_{i,t} = \delta \frac{W_t}{2} + (1 - \delta) v_{i,t}, \ i = 1, 2, \tag{8}$$

where we define $W_t = v_{1,t} + v_{2,t}$, $\delta = 1/(T_2 - 1)$, $\eta = \sqrt{8/T_2 \left(2 \ln 2 + (T_2 - 1) H(1/(T_2 - 1))\right)}$ and $H(x) = -x \ln x - (1 - x) \ln(1 - x)$ is the binary entropy function defined for $x \in (0, 1)$.

**Analysis** From rounds $t > T_1$, the first model $\mathbf{w}_{1,t}$ would become worse due to the cumulative recovered error while the second model will become better by the large amount of coming data. Since $\mathbf{w}_{1,t}$ is initialized by $\mathbf{w}_{1,T1}$ which is learnt from the old feature space and $\mathbf{w}_{2,t}$ is initialized randomly, it is reasonable to assume that $\mathbf{w}_{1,t}$ is better than $\mathbf{w}_{2,t}$ in the beginning, but inferior to $\mathbf{w}_{2,t}$ after sufficient large number of rounds. Let $s$ be the round after which $\mathbf{w}_{1,t}$ is worse than $\mathbf{w}_{2,t}$. We define $L^s = \sum_{t=T_1+1}^{s} \ell(f_{1,t}, y_t) + \sum_{t=s+1}^{T_2} \ell(f_{2,t}, y_t)$, we can verify that

$$\min_{T_1+1 \leq s \leq T_1+T_2} L^s \leq \min_{i=1,2} L^{S_i}. \tag{9}$$

Then a more ambitious goal is to compare the proposed algorithm against $\mathbf{w}_{1,t}$ from rounds $T_1 + 1$ to $s$, and against the $\mathbf{w}_{2,t}$ from rounds $s$ to $T_1 + T_2$, which motivates us to study the following performance measure $L^{S_{12}} - L^s$. Because the exact value of $s$ is generally unknown, we need to bound the worst-case $L^{S_{12}} - \min_{T_1+1 \leq s \leq T_1+T_2} L^s$. An upper bound of $L^{S_{12}}$ is given as follows.

**Theorem 2.** *For all $T_2 > 1$, if the model is run with parameter $\delta = 1/(T_2 - 1)$ and $\eta = \sqrt{8/T_2 \left(2 \ln 2 + (T_2 - 1) H(1/T_2 - 1)\right)}$, then*

$$L^{S_{12}} \leq \min_{T_1+1 \leq s \leq T_1+T_2} L^s + \sqrt{\frac{T_2}{2} \left(2 \ln 2 + \frac{H(\delta)}{\delta}\right)} \tag{10}$$

*where $H(x) = -x \ln x - (1 - x) \ln(1 - x)$ is the binary entropy function.*

Table 1: Detail description of datasets: let $n$ be the number of examples, and $d_1$ and $d_2$ denote the dimensionality of the first and second feature space, respectively. The first 9 datasets in the left column are synthetic datasets, "r.EN-GR" means the dataset EN-GR comes from Reuter and "RFID" is the real dataset.

| Dataset | n | $d_1$ | $d_2$ | Dataset | $n$ | $d_1$ | $d_2$ | Dataset | $n$ | $d_1$ | $d_2$ |
|---------|-----|-------|-------|---------|--------|--------|--------|---------|--------|--------|--------|
| Australian | 690 | 42 | 29 | r.EN-FR | 18,758 | 21,531 | 24,892 | r.GR-IT | 29,953 | 34,279 | 15,505 |
| Credit-a | 653 | 15 | 10 | r.EN-GR | 18,758 | 21,531 | 34,215 | r.GR-SP | 29,953 | 34,279 | 11,547 |
| Credit-g | 1,000 | 20 | 14 | r.EN-IT | 18,758 | 21,531 | 15,506 | r.IT-EN | 24,039 | 15,506 | 21,517 |
| Diabetes | 768 | 8 | 5 | r.EN-SP | 18,758 | 21,531 | 11,547 | r.IT-FR | 24,039 | 15,506 | 24,892 |
| DNA | 940 | 180 | 125 | r.FR-EN | 26,648 | 24,893 | 21,531 | r.IT-GR | 24,039 | 15,506 | 34,278 |
| German | 1,000 | 59 | 41 | r.FR-GR | 26,648 | 24,893 | 34,287 | r.IT-SP | 24,039 | 15,506 | 11,547 |
| Kr-vs-kp | 3,196 | 36 | 25 | r.FR-IT | 26,648 | 24,893 | 15,503 | r.SP-EN | 12,342 | 11,547 | 21,530 |
| Splice | 3,175 | 60 | 42 | r.FR-SP | 26,648 | 24,893 | 11,547 | r.SP-FR | 12,342 | 11,547 | 24,892 |
| Svmguide3 | 1,284 | 22 | 15 | r.GR-EN | 29,953 | 34,279 | 21,531 | r.SP-GR | 12,342 | 11,547 | 34,262 |
| RFID | 940 | 78 | 72 | r.GR-FR | 29,953 | 34,279 | 24,892 | r.SP-IT | 12,342 | 11,547 | 15,500 |

According to Theorem 2 we know that $L^{S_{12}}$ is comparable to $\min_{T_1+1 \leq s \leq T_1+T_2} L^s$. Due to (9), we can conclude that the upper bound of $L^{S_{12}}$ in Algorithm 3 is tighter than that of Algorithm 2.

# 5  Experiments

In this section, we first introduce the datasets we use. We want to emphasize that we collected one real dataset by ourselves since our setting of feature evolving is relatively novel so that the required datasets are not widely available yet. Then we introduce the compared methods and settings. Finally experiment results are given.

## 5.1  Datasets

We conduct our experiments on 30 datasets consisting of 9 synthetic datasets, 20 Reuter datasets and 1 real dataset. To generate synthetic data, we randomly choose some datasets from different domains including *economy* and *biology*, etc[1] whose scales vary from 690 to 3,196. They only have one feature space at first. We artificially map the original datasets into another feature space by random Gaussian matrices, then we have data both from feature space $S_1$ and $S_2$. Since the original data are in batch mode, we manually make them come sequentially. In this way, synthetic data are completely generated. We also conduct our experiments on 20 datasets from Reuter [3]. They are multi-view datasets which have large scale varying from 12,342 to 29,963. Each dataset has two views which represent two different kinds of languages, respectively. We regard the two views as the two feature spaces. Now they do have two feature spaces but the original data are in batch mode, so we will artificially make them come in streaming way.

We use the RFID technique to collect the real data which contain 450 instances from $S_1$ and $S_2$ respectively. RFID technique is widely used to do moving goods detection [31]. In our case, we want to utilize the RFID technique to predict the location's coordinate of the moving goods attached by RFID tags. Concretely, we arranged several RFID aerials around the indoor area. In each round, each RFID aerial received the tag signals, then the goods with tag moved, at the same time, we recorded the goods' coordinate. Before the aerials expired, we arranged new aerials beside the old ones to avoid the situation without aerials. So in this overlapping period, we have data from both old and new feature spaces. After the old aerials expired, we continue to use the new ones to receive signals. Then we only have data from feature space $S_2$. So the RFID data we collect totally satisfy our assumptions. The details of all the datasets we use are presented in Table 1.

## 5.2  Compared Approaches and Settings

We compare our FESL-c and FESL-s with three approaches. One is mentioned in Section 3, where once the feature space changed, the online gradient descent algorithm will be invoked from scratch, named as NOGD (Naive Online Gradient Descent). The other two approaches utilize the model learned from feature space $S_1$ by online gradient descent to do predictions on the recovered data. The

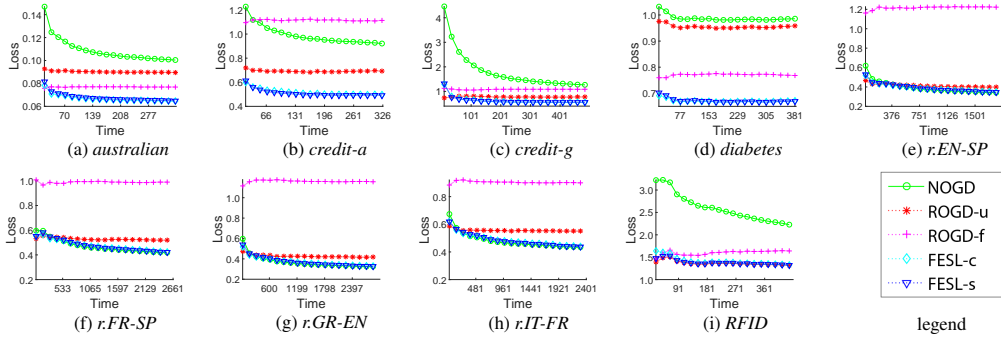

Figure 3: The trend of loss with three baseline methods and the proposed methods on synthetic data. The smaller the cumulative loss, the better. All the average cumulative loss at any time of our methods is comparable to the best of baseline methods and 8 of 9 are smaller.

difference between them is that one keeps updating with the recovered data while the other does not. The one which keeps updating is called Updating Recovered Online Gradient Descent (ROGD-u) and the other which keeps fixed is called Fixed Recovered Online Gradient Descent (ROGD-f). We evaluate the empirical performances of the proposed approaches on classification and regression tasks on rounds $T_1 + 1, \ldots, T_1 + T_2$. To verify that our analysis is reasonable, we present the trend of average cumulative loss. Concretely, at each time $t'$, the loss $\bar{\ell}_{t'}$ of every method is the average of the cumulative loss over $1, \ldots, t'$, namely $\bar{\ell}_{t'} = (1/t') \sum_{t=1}^{t'} \ell_t$. We also present the classification performance over all instances on rounds $T_1 + 1, \ldots, T_1 + T_2$ on synthetic and Reuter data. The performances of all approaches are obtained by average results over 10 independent runs on synthetic data. Due to the large scale of Reuter data, we only conduct 3 independent runs on Reuter data and report the average results.

The parameters we need to set are the number of instances in overlapping period, i.e., $B$, the number of instances in $S_1$ and $S_2$, i.e., $T_1$ and $T_2$ and the step size, i.e., $\tau_t$ where $t$ is time. For all baseline methods and our methods, the parameters are the same. In our experiments, we set $B$ 5 or 10 for synthetic data, 50 for Reuter data and 40 for RFID data. We set almost $T_1$ and $T_2$ to be half of the number of instances, and $\tau_t$ to be $1/(c\sqrt{t})$ where $c$ is searched in the range $\{1, 10, 50, 100, 150\}$. The detailed setting of $c$ in $\tau_t$ for each dataset is presented in supplementary file.

### 5.3 Results

Here we only present part of the loss trend results, and other results are presented in the supplementary file. Figure 3 gives the trend of average cumulative loss. (a-d) are the results on synthetic data, (e-h) are the results on Reuter data, (i) is the result of the real data. The smaller the average cumulative loss, the better. From the experimental results, we have the following observations. First, all the curves with circle marks representing NOGD decrease rapidly which conforms to the fact that NOGD on rounds $T_1 + 1, \ldots, T_1 + T_2$ becomes better and better with more and more data coming. Besides, the curves with star marks representing ROGD-u also decline but not very apparent since on rounds $1, \ldots, T_1$, ROGD-u already learned well and tend to converge, so updating with more recovered data could not bring too much benefits. Moreover, the curves with plus marks representing ROGD-f does not drop down but even go up instead, which is also reasonable because it is fixed and if there are some recovering errors, it will perform worse. Lastly, our methods are based on NOGD and ROGD-u, so their average cumulative losses also decrease. As can be seen from Figure 3, the average cumulative losses of our methods are comparable to the best of baseline methods on all datasets and are smaller than them on 8 datasets. And FESL-s exhibits slightly smaller average cumulative loss than FESL-c. You may notice that NOGD is always worse than ROGD-u on synthetic data and real data while on Reuter data NOGD becomes better than ROGD-u after a few rounds. This is because on synthetic data and real data, we do not have enough rounds to let all methods converge while on Reuter data, large amounts of instances ensure the convergence of every method. So when all the methods converge, we can see that NOGD is better than other baseline methods since it always receives the real instances while ROGD-u and ROGD-f receive the recovered instances which may contain recovered error. As can be seen from (e-h), in the first few rounds, our methods are comparable to ROGD-u. When NOGD is better than ROGD-u, our methods are comparable to NOGD which shows that our methods

Table 2: Accuracy with its variance on synthetic datasets and Reuter datasets. The larger the better. The best ones among all the methods are bold.

| Dataset | NOGD | ROGD-u | ROGD-f | FESL-c | FESL-s |
|---|---|---|---|---|---|
| australian | .767±.009 | **.849±.009** | .809±.025 | **.849±.009** | **.849±.009** |
| credit-a | .811±.006 | .826±.018 | .785±.051 | .827±.014 | **.831±.009** |
| credit-g | .659±.010 | **.733±.006** | .716±.011 | **.733±.006** | **.733±.006** |
| diabetes | .650±.002 | **.652±.009** | .651±.006 | **.652±.007** | **.652±.009** |
| dna | .610±.013 | .691±.023 | .608±.064 | .691±.023 | **.692±.021** |
| german | .684±.006 | .700±.002 | .700±.002 | .700±.001 | **.703±.004** |
| kr-vs-kp | .612±.005 | .621±.036 | .538±.024 | .626±.028 | **.630±.016** |
| splice | .568±.005 | **.612±.022** | .567±.057 | **.612±.022** | **.612±.022** |
| svmguide3 | .680±.010 | **.779±.010** | .748±.012 | **.779±.010** | .778±.010 |
| r.EN-FR | .902±.004 | .849±.003 | .769±.069 | **.903±.003** | .902±.005 |
| r.EN-GR | .867±.005 | .836±.007 | .802±.036 | **.870±.002** | **.870±.003** |
| r.EN-IT | .858±.014 | .847±.014 | .831±.018 | .861±.010 | **.863±.013** |
| r.EN-SP | .900±.002 | .848±.002 | .825±.001 | **.901±.001** | .899±.002 |
| r.FR-EN | **.858±.007** | .776±.009 | .754±.012 | **.858±.007** | **.858±.007** |
| r.FR-GR | .869±.004 | .774±.019 | .753±.021 | **.870±.004** | .868±.003 |
| r.FR-IT | .874±.005 | .780±.022 | .744±.040 | **.874±.005** | .873±.005 |
| r.FR-SP | **.872±.001** | .778±.022 | .735±.013 | **.872±.001** | .871±.002 |
| r.GR-EN | **.907±.000** | .850±.007 | .801±.035 | **.907±.001** | .906±.000 |
| r.GR-FR | **.898±.001** | .827±.009 | .802±.023 | **.898±.001** | **.898±.000** |
| r.GR-IT | .847±.011 | **.851±.017** | .816±.006 | .850±.018 | **.851±.017** |
| r.GR-SP | **.902±.001** | .845±.003 | .797±.012 | **.902±.001** | **.902±.001** |
| r.IT-EN | .854±.003 | .760±.006 | .730±.024 | **.856±.002** | .854±.003 |
| r.IT-FR | .863±.002 | .753±.012 | .730±.020 | **.864±.002** | .862±.003 |
| r.IT-GR | .849±.004 | .736±.022 | .702±.012 | **.849±.004** | .846±.004 |
| r.IT-SP | **.839±.006** | .753±.014 | .726±.005 | **.839±.007** | **.839±.006** |
| r.SP-EN | **.926±.002** | .860±.005 | .814±.021 | **.926±.002** | .924±.001 |
| r.SP-FR | .876±.005 | .873±.017 | .833±.042 | .876±.014 | **.878±.012** |
| r.SP-GR | .871±.013 | .827±.025 | .810±.026 | **.873±.013** | **.873±.013** |
| r.SP-IT | **.928±.002** | .861±.005 | .826±.005 | **.928±.003** | .927±.002 |

are comparable to the best one all the time. Moreover, FESL-s performs worse than FESL-c in the beginning while afterwards, it becomes slightly better than FESL-c.

Table 2 shows the accuracy results on synthetic datasets and Reuter datasets. We can see that for synthetic datasets, FESL-s outperforms other methods on 8 datasets, FESL-c gets the best on 5 datasets and ROGD-u also gets 5. NOGD performs worst since it starts from scratch. ROGD-u is better than NOGD and ROGD-f because ROGD-u exploits the old better trained model from old feature space and keep updating with recovered instances. Our two methods are based on NOGD and ROGD-u. We can see that our methods can follow the best baseline method or even outperform it. For Reuter datasets, we can see that FESL-c outperforms other methods on 17 datasets, FESL-s gets the best on 9 datasets and NOGD gets 8 while ROGD-u gets 1. In Reuter datasets, the period on new feature space is longer than that in synthetic datasets so that NOGD can update itself to a good model. Whereas ROGD-u updates itself with recovered data, so the model will become worse when recovered error accumulates. ROGD-f does not update itself, thus it performs worst. Our two methods can take the advantage of NOGD and ROGD-f and perform better than them.

# 6 Conclusion

In this paper, we focus on a new setting: feature evolvable streaming learning. Our key observation is that in learning with streaming data, old features could vanish and new ones could occur. To make the problem tractable, we assume there is an overlapping period that contains samples from both feature spaces. Then, we learn a mapping from new features to old features, and in this way both the new and old models can be used for prediction. In our first approach FESL-c, we ensemble two predictions by learning weights adaptively. Theoretical results show that the assistance of the old feature space can improve the performance of learning with streaming data. Furthermore, we propose FESL-s to dynamically select the best model with better performance guarantee.

**Acknowledgement** This research was supported by NSFC (61333014, 61603177), JiangsuSF (BK20160658), Huawei Fund (YBN2017030027) and Collaborative Innovation Center of Novel Software Technology and Industrialization.

## Footnotes

[1] Datasets can be found in http://archive.ics.uci.edu/ml/.

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
