[Supplementary Material]

# Supplementary Material of "Learning with Feature Evolvable Streams"

**Bo-Jian Hou    Lijun Zhang    Zhi-Hua Zhou**
National Key Laboratory for Novel Software Technology,
Nanjing University, Nanjing, 210023, China
{houbj,zhanglj,zhouzh}@lamda.nju.edu.cn

In the supplementary material, we will prove the two theorems in the section "Our Proposed Approaches", and give some additional experiment results with the detailed setting of step size $\tau_t$.

## 1  Analysis

In this section, we will give the detailed proofs of the two theorems in "Our Proposed Approaches". The two theorems are the special cases of Theorem 2.2 and Corollary 5.1 respectively in [1].

### 1.1  Proof of Theorem 1

At first, we restate Theorem 1 as follows:

**Theorem 1.** *Assume that the loss function $\ell$ is convex in its first argument and that it takes value in [0,1]. For all $T_2 > 1$ and for all $y_t \in \mathcal{Y}$ with $t = T_1 + 1, \ldots, T_1 + T_2$, $L^{S_{12}}$ with parameter $\eta_t = \sqrt{8(\ln 2)/T_2}$ satisfies*

$$L^{S_{12}} \leq \min(L^{S_1}, L^{S_2}) + \sqrt{(T_2/2)\ln 2}. \tag{1}$$

To prove Theorem 1, we propose to bound the related quantities $(1/\eta)\ln(A_t/A_{t-1})$ where

$$A_t = \sum_{i=1}^{2} \alpha_{i,t} = \sum_{i=1}^{2} e^{-\eta L_t^{S_i}}$$

for $t \geq T_1$, and $A_{T_1} = 2$. $L_t^{S_i}$ is the cumulative loss at time $t$ of the $i$-th base learner, namely $L_t^{S_i} = \sum_{s=T_1+1}^{t} \ell(f_{i,s}, y_s)$. In the proof we use the following classical inequality due to Hoeffding [2].

**Lemma 1.** *Let $X$ be a random variable with $a \leq X \leq b$. Then for any $s \in \mathbb{R}$,*

$$\ln \mathbb{E}[e^{sX}] \leq s\mathbb{E}X + \frac{s^2(b-a)^2}{8}$$

The detailed proof of Lemma 1 can be found in Section A.1 of the Appendix in [1].

*Proof of Theorem 1.*  First observe that

$$
\begin{aligned}
\ln \frac{A_{T_1+T_2}}{A_{T_1}} &= \ln\left(\sum_{i=1}^{2} e^{-\eta L_{T_1+T_2}^{S_i}}\right) - \ln 2 \\
&\geq \ln\left(\max_{i=1,2} e^{-\eta L_{T_1+T_2}^{S_i}}\right) - \ln 2 \\
&= -\eta \min_{i=1,2} L_{T_1+T_2}^{S_i} - \ln 2.
\end{aligned}
\tag{2}
$$

On the other hand, for each $t = T_1 + 1, \ldots, T_1 + T_2$,

$$\ln \frac{A_t}{A_{t-1}} = \ln \frac{\sum_{i=1}^2 e^{-\eta \ell(f_{i,t}, y_t)} e^{-\eta L_{t-1}^{S_i}}}{\sum_{j=1}^2 e^{-\eta L_{t-1}^{S_j}}}$$

$$= \ln \frac{\sum_{i=1}^2 \alpha_{i,t-1} e^{-\eta \ell(f_{i,t}, y_t)}}{\sum_{j=1}^2 \alpha_{j,t-1}}.$$

Now using Lemma 1, we observe that the quantity above may be upper bounded by

$$-\eta \frac{\sum_{i=1}^2 \alpha_{i,t-1} \ell(f_{i,t}, y_t)}{\sum_{j=1}^2 \alpha_{j,t-1}} + \frac{\eta^2}{8}$$

$$\leq -\eta \ell \left( \frac{\sum_{i=1}^2 \alpha_{i,t-1} f_{i,t}}{\sum_{j=1}^2 \alpha_{j,t-1}}, y_t \right) + \frac{\eta^2}{8}$$

$$= -\eta \ell(\widehat{p}_t, y_t) + \frac{\eta^2}{8}$$

where we used the convexity of the loss function in its first argument and the way how the weight updates. Summing over $t = T_1 + 1, \ldots, T_1 + T_2$, we get

$$\ln \frac{A_{T_1+T_2}}{A_{T_1}} \leq -\eta L^{S_{12}} + \frac{\eta^2}{8} T_2. \tag{3}$$

Combining this with the lower bound (2) and solving for $L^{S_{12}}$, we find that

$$L^{S_{12}} \leq \min(L^{S_1}, L^{S_2}) + \frac{\ln 2}{\eta} + \frac{\eta}{8} T_2$$

as desired. In particular, with $\eta = \sqrt{8 \ln 2 / T_2}$, the upper bound becomes $\min(L^{S_1}, L^{S_2}) + \sqrt{(T_2/2) \ln 2}$. $\qquad \square$

## 1.2 Proof of Theorem 2

The Theorem 2 in our paper is restated as follows:

**Theorem 2.** *For all $T_2 > 1$, if the model is run with parameter $\delta = 1/(T_2 - 1)$ and $\eta = \sqrt{8/T_2 (2 \ln 2 + (T_2 - 1) H(1/T_2 - 1))}$, then*

$$L^{S_{12}} \leq \min_{T_1+1 \leq s \leq T_1+T_2} L^s + \sqrt{\frac{T_2}{2} \left( 2 \ln 2 + (T_2 - 1) H(\frac{1}{T_2 - 1}) \right)} \tag{4}$$

*where $H(x) = -x \ln x - (1-x) \ln(1-x)$ is the binary entropy function.*

To prove Theorem 2, we first give some definitions. Since we only choose one base learner's prediction in FESL-s as our final prediction in each round, we use $I_t \in \{1, 2\}$ to denote the index of the base learners in $t$-th round for $t = T_1 + 1, \ldots, T_1 + T_2$. We call $I_t$ an action. So the loss in round $t$ can be denoted as $\ell(I_t, y_t)$. Thus, randomly choosing one base learner in each round is a randomized version of FESL-c, so we call it randomized FESL-c. Denote the distribution according to which the random action $I_t$ is drawn at time $t$ by $\boldsymbol{p}_t = (p_{1,t}, p_{2,t})$, and $\bar{\ell}(\boldsymbol{p}_t, y_t) = \sum_{i=1}^2 p_{i,t} \ell(I_t, y_t)$ is the expected loss of randomized FESL-c at time $t$. Then we have the following lemma:

**Lemma 2.** *Let $T_2 > 1$ and $\delta \in (0, 1)$. The randomized FESL-c with $\eta = \sqrt{8 \ln 2 / n}$ satisfies, with probability at least $1 - \delta$*

$$\sum_{t=T_1+1}^{T_1+T_2} \ell(I_t, y_t) - \min_{i=1,2} \sum_{t=T_1+1}^{T_1+T_2} \ell(i, y_t) \leq \sqrt{\frac{T_2 \ln 2}{2}} + \sqrt{\frac{T_2}{2} \ln \frac{1}{\delta}}.$$

*Proof.* The random variables $\ell(I_t, y_t) - \bar{\ell}(\boldsymbol{p}_t, y_t)$, for $t = T_1 + 1 \ldots, T_1 + T_2$, form a sequence of bounded martingale differences. With a simple application of the Hoeffding-Azuma inequality and combining the results of Theorem 1, we yield the result of this lemma. $\qquad \square$

In addition, $i_{T_1+1}, \ldots, i_s, i_{s+1}, \ldots, i_{T_1+T_2}$ is defined as the sequence of the base learner's index such that we can study a more ambitious goal $g = L^{S_{12}} - L^s$ where $L^s = \sum_{t=T_1+1}^{T_1+T_2} \ell(i_t, y_t)$. It is not difficult to modify the randomized FESL-c in order to achieve this goal. Specifically, we associate a *compound action* with each sequence which only switches once. Then we can run our randomized FESL-c over the set of compound actions: at any time $t$ the randomized FESL-c draws a compound action $(I_{T_1+1}, \ldots, I_{T_1+T_2})$ and plays action $I_t$. Denote by $M$ the number of all compound actions. Then, in FESL-c, we only have 2 base learners while in randomized FESL-c, we have $M$ base learners. Then Lemma 2 implies that $g$ is bounded by $\sqrt{(T_2 \ln M)/2}$. Hence, it suffices to count the number of compound actions: for each $k = 0, \ldots, 1$ there are $C_{T_2-1}^k$ ways to pick $k$ time steps $t = T_1 + 1, \ldots, T_1 + T_2 - 1$ where a switch $i_t \neq i_{t+1}$ occurs, and there are $2(2-1)^k$ ways to assign a distinct action to each of the $k+1$ resulting blocks. This gives

$$M = \sum_{k=0}^{m} C_{T_2-1}^k 2 \leq 4 \exp\left((T_2 - 1)H\left(\frac{1}{T_2 - 1}\right)\right).$$

where $H(x) = -x \ln x - (1-x)\ln(1-x)$ is the binary entropy function defined for $x \in (0,1)$. Substituting this bound in $\sqrt{(T_2 \ln M)/2}$, we find that $g$ satisfies

$$g \leq \sqrt{\frac{T_2}{2}\left(2 \ln 2 + (T_2 - 1)H(\frac{1}{T_2 - 1})\right)}$$

on any action sequence $i_{T_1+1}, \ldots, i_s, i_{s+1}, \ldots, i_{T_1+T_2}$. However, the randomized FESL-c requires to explicitly manage an exponential number of compound actions in its straightforward implementation. Then we propose FESL-s which can efficiently implement a generalized version of randomized FESL-c that is able to achieve $g$. Specifically, FESL-s is derived from a variant of randomized FESL-c where the initial weight distribution is not uniform. We have the following results.

**Lemma 3.** *For all $T_2 > 1$, if the randomized FESL-c is run using initial weights $\alpha_{1,T_1}, \alpha_{2,T_1} \geq 0$ such that $A_{T_1+T_2} = \alpha_{1,T_1+T_2} + \alpha_{2,T_1+T_2} \leq 1$, then*

$$\sum_{t=T_1+1}^{T_1+T_2} \bar{\ell}(\boldsymbol{p}_t, y_t) \leq \frac{1}{\eta} \ln \frac{1}{A_{T_1+T_2}} + \frac{\eta}{8}T_2,$$

*where*

$$A_{T_1+T_2} = \sum_{i=1}^{2} \alpha_{i,T_1+T_2} = \sum_{i=1}^{2} \alpha_{i,T_1} e^{-\eta \sum_{t=T_1+1}^{T_1+T_2} \ell(i, y_t)}$$

*is the sum of the weights after $T_2$ rounds.*

*Proof.* From equation (3) mentioned in the last subsection, we know that

$$\ln \frac{A_{T_1+T_2}}{A_{T_1}} \leq -\eta \sum_{t=T_1}^{T_1+T_2} \bar{\ell}(\boldsymbol{p}_t, y_t) + \frac{\eta^2}{8}T_2$$

where $A_t = \sum_{i=1}^{2} \alpha_{i,t} = \sum_{i=1}^{2} e^{-\eta L_t^{S_i}}$. Since $A_{T_1} \leq 1$, then we have

$$\begin{aligned}
\sum_{t=T_1+1}^{T_1+T_2} \bar{\ell}(\boldsymbol{p}_t, y_t) &\leq \frac{1}{\eta} \ln A_{T_1} - \frac{1}{\eta} \ln A_{T_1+T_2} + \frac{\eta T_2}{8} \\
&= \frac{1}{\eta} \ln \frac{1}{A_{T_1+T_2}} + \frac{\eta T_2}{8} - \frac{1}{\eta} \ln \frac{1}{A_{T_1}} \\
&\leq \frac{1}{\eta} \ln \frac{1}{A_{T_1+T_2}} + \frac{\eta T_2}{8}.
\end{aligned}$$

$\square$

We write $\alpha'_t(i_{T_1+1}, \ldots, i_{T_1+T_2})$ to denote the weight assigned at time $t$ by the randomized FESL-c to the compound action $(i_{T_1+1}, \ldots, i_{T_1+T_2})$. For any fixed choice of the parameter $\delta \in (0, 1)$, the initial weights of the compound actions are defined by

$$\alpha'_{T_1}(i_{T_1+1}, \ldots, i_{T_1+T_2}) = \frac{1}{2} \left(\frac{\delta}{2}\right) \left(1 - \delta + \frac{\delta}{2}\right)^{T_2-1}.$$

Then the way of updating weight is as follows:

$$\alpha'_t(i_{T_1+1}, \ldots, i_{T_1+T_2}) = \alpha'_{T_1}(i_{T_1+1}, \ldots, i_{T_1+T_2}) \exp\left(-\eta \sum_{s=1}^{t} \ell(i_s, y_s)\right).$$

Introducing the "marginalized" weights

$$\alpha'_{T_1}(i_{T_1+1}, \ldots, i_{T_1+T_2}) = \sum_{i_{t+1}, \ldots, i_{T_1+T_2}} \alpha'_{T_1}(i_{T_1+1}, \ldots, i_t, i_{t+1}, \ldots, i_{T_1+T_2})$$

for all $t = T_1 + 1, \ldots, T_1 + T_2$, we obtain that FESL-s draws action $i$ at time $t + 1$ with probability $\alpha'_{i,t}/A'_t$, where $A'_t = \alpha'_{1,t} + \alpha'_{2,t}$ and

$$\alpha'_{i,t} = \sum_{i_1, \ldots, i_t, i_{t+2}, \ldots, i_n} \alpha'_t(i_{T_1+1}, \ldots, i_t, i, i_{t+2}, \ldots, i_{T_1+T_2})$$

for $t \geq T_1 + 1$ and $\alpha'_{i,T_1} = 1/2$.

The initial weights are recursively computed as follows

$$\alpha'_{T_1}(i_1) = 1/2,$$
$$\alpha'_{T_1}(i_{T_1+1}, \ldots, i_{t+1}) = \alpha'_{T_1}(i_{T_1+1}, \ldots, i_t) \left(\frac{\delta}{2} + (1 - \delta)\mathbb{I}_{\{i_{t+1}=i_t\}}\right).$$

The following result shows that FESL-s is indeed an efficient version of randomized FESL-c.

**Theorem 3.** *For all $i = 1, 2, t = T_1 + 1, \ldots, T_1 + T_2, \delta \in [0, 1]$, we have $\alpha_{i,t} = \alpha'_{i,t}$, where $\alpha_{i,t}$ is the weight of the $i$-th base learner at time $t$ in FESL-s, and $\alpha'_{i,t}$ is the weight of the conditional distribution of action $I'_t$ drawn at time $t$ by randomized FESL-c run over the compound actions $(i_{T_1+1}, \ldots, i_{T_1+T_2})$ using initial weights $\alpha'_{T_1}(i_{T_1+1}, \ldots, i_{T_1+T_2})$ set with the same value of $\delta$.*

*Proof.* We proceed by induction on $t$. For $t = T_1$, $\alpha_{i,T_1} = \alpha'_{i,T_1} = 1/2$ for all $i$. For the induction step, assume that $\alpha_{i,s} = \alpha'_{i,s}$ for all $i$ and $s < t$. We have

$$\alpha'_{i,t} = \sum_{i_1,\ldots,i_t,i_{t+2},\ldots,i_n} \alpha'_t(i_{T_1+1},\ldots,i_t,i,i_{t+2},\ldots,i_{T_1+T_2})$$

$$= \sum_{i_{T_1+1},\ldots,i_t} \exp\left(-\eta\sum_{s=1}^t \ell(i_s,y_s)\right) \alpha'_{T_1}(i_{T_1+1},\ldots,i_t,i)$$

$$= \sum_{i_{T_1+1},\ldots,i_t} \exp\left(-\eta\sum_{s=1}^t \ell(i_s,y_s)\right) \alpha'_{T_1}(i_{T_1+1},\ldots,i_t)\frac{\alpha'_{T_1}(i_{T_1+1},\ldots,i_t,i)}{\alpha'_{T_1}(i_{T_1+1},\ldots,i_t)}$$

$$= \sum_{i_{T_1+1},\ldots,i_t} \exp\left(-\eta\sum_{s=1}^t \ell(i_s,y_s)\right) \alpha'_{T_1}(i_{T_1+1},\ldots,i_t)\left(\frac{\delta}{2} + (1-\delta)\mathbb{I}_{\{i_t=i\}}\right)$$

(using the recursive definition of $\alpha'_{T_1}$)

$$= \sum_{i_t} e^{-\eta\ell(i_t,y_t)}\alpha'_{i_t,t-1}\left(\frac{\delta}{2} + (1-\delta)\mathbb{I}_{\{i_t=i\}}\right)$$

$$= \sum_{i_t} e^{-\eta\ell(i_t,y_t)}\alpha_{i_t,t-1}\left(\frac{\delta}{2} + (1-\delta)\mathbb{I}_{\{i_t=i\}}\right)$$

(by the induction hypothesis)

$$= \sum_{i_t} v_{i_t,t}\left(\frac{\delta}{2} + (1-\delta)\mathbb{I}_{\{i_t=i\}}\right)$$

(using (9).1 from "Dynamic Selection")

$$= \alpha_{i,t} \quad \text{(using (9).2 from "Dynamic Selection")}$$

$\square$

Then we have a general result for FESL-s.

**Theorem 4.** *For all $n \geq T_1 + 1$, the goal of the FESL-s g satisfies*

$$g = \sum_{t=T_1+1}^n \bar{\ell}(\boldsymbol{p}_t,y_t) - \sum_{t=T_1+1}^n \ell(i_t,y_t) \leq \frac{2}{\eta}\ln 2 + \frac{1}{\eta}\ln\frac{1}{(\delta/2)(1-\delta)^{n-2}} + \frac{\eta}{8}n$$

*for all action sequences $i_{T_1+1},\ldots,i_{T_1+T_2}$.*

*Proof.* For a compound action $i_{T_1+1},\ldots,i_{T_1+T_2}$ we have

$$\ln\alpha'_{T_1+T_2}(i_{T_1+1},\ldots,i_{T_1+T_2}) = \ln\alpha'_{T_1}(i_{T_1+1},\ldots,i_{T_1+T_2}) - \eta\sum_{t=T_1+1}^{T_1+T_2} \ell(i_t,y_t).$$

By definition of $\alpha'_{T_1}$,

$$\alpha'_{T_1}(i_{T_1+1},\ldots,i_{T_1+T_2}) = \frac{1}{N}\left(\frac{\delta}{2}\right)\left(\frac{\delta}{2} + (1-\delta)\right)^{T_1+T_2-2} \geq \frac{1}{2}\left(\frac{\delta}{2}\right)(1-\delta)^{T_1+T_2-2}.$$

Therefore, using this in the bound of Lemma 3 we get, for any sequence $(i_{T_1+1},\ldots,i_{T_1+T_2})$,

$$\sum_{t=1}^n \bar{\ell}(\boldsymbol{p}_t,y_t) \leq \frac{1}{\eta}\ln\frac{1}{A'_{T_1+T_2}} + \frac{\eta}{8}T_2$$

$$\leq \frac{1}{\eta}\ln\frac{1}{\alpha'_{T_1+T_2}(i_{T_1+1},\ldots,i_{T_1+T_2})} + \frac{\eta}{8}T_2$$

$$\leq \sum_{t=1}^n \ell(i_t,y_t) + \frac{1}{\eta}\ln 2 + \frac{1}{\eta}\ln\frac{2}{\delta} - \frac{T_2-2}{\eta}\ln(1-\delta) + \frac{\eta}{8}T_2,$$

which concludes the proof. $\square$

(a) *dna*  (b) *german*  (c) *kr-vs-kp*

(d) *splice*  (e) *svmguide3*  legend

Figure 1: The trend of loss with three baseline methods and the proposed methods on synthetic data. The smaller the cumulative loss, the better.

With Lemma 3 and Theorem 4, we give the proof of Theorem 2 as follows.

*Proof of Theorem 2.* First, note that for $\delta = 1/(T_2 - 1)$

$$\ln \frac{1}{\delta(1-\delta)^{T_2-2}} = -\ln \frac{1}{T_2 - 1} - (T_2 - 2)\ln \frac{T_2 - 2}{T_2 - 1} = (T_2 - 1)H(\frac{1}{T_2 - 1}).$$

Using

$$\eta = \sqrt{\frac{8}{T_2}\left(2\ln 2 + (T_2 - 1)H(\frac{1}{T_2 - 1})\right)}$$

in the bound of Theorem 4 we obtain that

$$\sum_{t=T_1+1}^{T_1+T_2} \bar{\ell}(\boldsymbol{p}_t, y_t) - \sum_{t=T_1+1}^{T_1+T_2} \ell(i_t, y_t) \leq \sqrt{\frac{T_2}{2}\left(2\ln 2 + (T_2 - 1)H(\frac{1}{T_2 - 1})\right)}$$

for all action sequences $i_{T_1+1}, \ldots, i_{T_1+T_2}$, namely,

$$L^{S_{12}} \leq \min_{T_1+1 \leq s \leq T_1+T_2} L^s + \sqrt{\frac{T_2}{2}\left(2\ln 2 + \frac{H(\delta)}{\delta}\right)}$$

$\square$

## 2 Additional Experiments

In this section, the remaining loss trend results of 5 synthetic datasets and 16 results of Reuter datasets are presented. We also show the detailed setting of step size $\tau_t$ for each datasets.

As can be seen from Figure 1, the average cumulative loss of our methods is comparable to the best of baseline methods on all datasets and. And FESL-s exhibits slightly smaller average cumulative loss than FESL-c. We can also see from Figure 2 that, the average cumulative loss at any time of our methods is comparable to the best of baseline methods. Specifically, at first, ROGD-u is better than NOGD and our methods is comparable to ROGD-u. Afterwards, with more and more data coming, NOGD becomes better, then our methods are comparable to NOGD. Moreover, FESL-s performs

Figure 2: The trend of loss with three baseline methods and the proposed methods on Reuter data. The smaller the cumulative loss is, the better. The average cumulative loss at any time of our methods is smaller than the best of baseline methods.

worse than FESL-c in the beginning while afterwards, it becomes slightly better than FESL-c. Lastly, ROGD-f always performs the worst among all the approaches.

In our experiments, we set the step size $\tau_t$ to be $1/(c\sqrt{t})$ where $c$ is searched in the range $\{1, 10, 50, 100, 150\}$. Concretely, for synthetic datasets, we set $c$

- 1 for *australian*, *credit-a*, *credit-g* and *svmguide3*;
- 10 for *diabetes* and *splice*;
- 50 for *german*;
- 100 for *kr-vs-kp*;
- 150 for *dna*.

For Reuter datasets, we set $c$

- 10 for *r.GR-IT*, *r.GR-SP*, *r.SP-FR*;
- 50 for *r.EN-FR*, *r.EN-IT*, *r.EN-SP*, *r.FR-GR*, *r.FR-IT*, *r.FR-SP*, *r.GR-EN*, *r.IT-EN*, *r.IT-FR*, *r.IT-GR*, *r.IT-SP*, *r.SP-EN*, *r.SP-IT*;
- 100 for *r.FR-EN*;
- 150 for *r.EN-GR*, *r.GR-FR*, *r.SP-GR*.