[Reviews · NeurIPS 2017]

Reviewer 1



This paper formalizes a new problem setting, Feature Evolvable Streaming Learning. Sensors or other devices to extract feature values have the limited lifespans; therefore, these devices have been periodically replaced and the associated feature space changes. This learning paradigm prepares the overlapping period to adapt to the new feature space. In this overlapping period, learning algorithms receive features from both the old devices and the new devices simultaneously to capture the relationship between two feature spaces. This paper develops two learning algorithms to efficiently use previous experiences extracted from old training data to train/predict in the new feature space: 1) the weighted combination based predictor ensemble method, 2) the dynamic classifier selection. These two algorithms are proven to have theoretical performance guarantees respectively compared with the naive methods. Derived algorithms are simple for overcoming the difficulty in the feature evolvable streaming setting. Overall, this paper is interesting and well-written. The motivation is clear. The discussion is easy to follow. Besides, experiments include the real data setting, in which the RFID technique is used to gather data. This experiment demonstrates the usefulness of the proposed FESL problem setting. I would like to ask authors about the effectiveness of combining different predictors is. In the real-data experiment, ROGD-u (a baseline method) is comparable to the proposed methods. This result implies that sophisticated feature transformations between two feature spaces are keys in the FESL setting, not how to combine two different predictors intelligently. We can see the combining two predictors have relatively small effects. Is there any good realistic example to show the intelligent combination of predictors works significantly better than the naive method? Is it necessary if we could improve the transformation method between two feature spaces? As a minor note, if my understanding is correct, ROGD-u is also the proposed method of this paper, although this paper introduced this method as the baseline method. I think authors should insist on the importance of feature transformation and introduce these ROGDs as the proposed methods even without theoretical guarantees.

Reviewer 2



This paper presents two new methods for learning with streaming data under the assumption that feature spaces may evolve so old features vanish and new features occur. Examples of this scenario are environmental monitoring sensors or RFIDs for moving goods detection. Two different approaches are proposed, both based on the combination of two models: one trained with the old features and the other one based on the new features. Depending on how these models are combined, the authors propose two different methods for Feature Evolvable Streaming Learning (FESL): FESL-c that combines the predictions of both models, and FESL-s that selects the best model each time. Theoretical bounds on the cumulative loss of the both FESL methods as a function of the cumulative loss of the two models trained over different feature spaces are also provided. Experiments on datasets from the UCI Machine Learning Repository, Reuter’s data, and RFID real data have been conducted, and results show that the proposed models are able to provide competitive results in all cases when compared to three baseline methods. Overall, I think this is a very interesting paper that addresses a real-world problem that is quite common, for example, in environmental sensoring applications. I think that the paper is well-written and well-organized, and the experiments show the significance of the proposed methodology. Additionally, I positively value the use of RFID real data. * Contribution with respect to transfer learning. My main concern is about the quantification of the degree of contribution of this work, as I cannot see clearly the difference what is the difference between the proposed approach and the use of transfer learning in two different feature sets (of course, taking into account that transfer learning cannot be applied from t=1,…, T1-B). I would like the authors to clarify this point. * Theorems 1 and 2. I would also like the authors to discuss about the practical implications of Theorem 1 and Theorem 2, as in both cases convergence to 0 is obtained when T2 is large enough but that is not the case in Figures 1 and 2. Additionally, it seems to me that although the upper bound for FESL-s algorithm is tighter than for the FESL-c method, the convergence is slower. Please, clarify this point. * Optimal value for s. I do not understand why there is an inequality in Equation 9 instead of an equality according to the assumptions made (the first model is better than the second one for t smaller than s, and the second model is better than the first one for t larger than s). In this regard, I can see in Figure 3 that this assumption holds in practice, but the value for s depends on the problem you are working on. Providing some intuition about the range of values for s would be valuable. * Datasets and experimental setup. I have some questions: - I would like the authors to clarify why the Gaussian noise was added to the UCI datasets instead of considering two different subsets of features. I guess that this is because it is assumed that the features spaces are not so different, but I would like the authors to discuss this point. - Please, provide more details about how batch datasets was transformed into streaming data. I suppose it was done randomly, but then I am wondering whether results on 10 independent rounds are enough to have reliable results and I would like to see the variability of the results provided in terms of the cumulative loss. - It is not clear to me how the hyperparameters of the methods (delta and eta) were adjusted. Were they set according to Theorems 1 and 2? What about the hyperparameters in the baseline methods? - I do not understand the sentence (page 7): “So for all baseline methods and our methods, we use the same parameters and the comparison is fair”. * Results. It would be desirable to provide similar tables to Table 1 for the Reuter’s and RFID data.

Reviewer 3



In this work, the authors propose a new setting in online learning area. They assume the old feature vould vanish and new feature could occur and there exist an overlapping period that contains both feature space. The authors propose a mapping strategy to recover the vanished feature and exploit it to improve performance. Besides, the authors propsoed two ensembele methhods with performance guarantee. The idea is interesting and straitforward, the paper is easy to follow. My major concerns are: 1. The key idea is similar with [15], both of them design a constraint between vanished feature and new feature. However, i think the assumption and solution of this work is a special case of feature evolvable streams problem, especially for the assumption about overlapping period. That is, the titile is not appropriate for this work. 2. The theoretical analysis is a directly extension of learning with expert advice [5,28]. Thus, the theoretical result is not novel.